# Dermatitis Herpetiformis: An Update on Diagnosis, Disease Monitoring, and Management

**DOI:** 10.3390/medicina57080843

**Published:** 2021-08-20

**Authors:** Christopher N. Nguyen, Soo-Jung Kim

**Affiliations:** 1School of Medicine, Baylor College of Medicine, Houston, TX 77030, USA; cnnguyen@bcm.edu; 2Department of Dermatology, Baylor College of Medicine, Houston, TX 77030, USA

**Keywords:** dermatitis herpetiformis, celiac disease, bullous, autoimmune, pruritis, disease monitoring

## Abstract

Dermatitis herpetiformis (DH), Duhring disease, is caused by gluten sensitivity and affects 11.2 to 75.3 per 100,000 people in the United States and Europe with an incidence of 0.4 to 3.5 per 100,000 people per year. DH is characterized by a symmetrical blistering rash on the extensor surfaces with severe pruritus. The diagnosis continues to be made primarily by pathognomonic findings on histopathology, especially direct immunofluorescence (DIF). Recently, anti-epidermal transglutaminase (TG3) antibodies have shown to be a primary diagnostic serology, while anti-tissue transglutaminase (TG2) and other autoantibodies may be used to support the diagnosis and for disease monitoring. Newly diagnosed patients with DH should be screened and assessed for associated diseases and complications. A gluten-free diet (GFD) and dapsone are still mainstays of treatment, but other medications may be necessary for recalcitrant cases. Well-controlled DH patients, managed by a dermatologist, a gastroenterologist, and a dietician, have an excellent prognosis. Our review comprehensively details the current diagnostic methods, as well as methods used to monitor its disease course. We also describe both the traditional and novel management options reported in the literature.

## 1. Introduction

Dermatitis herpetiformis (DH) is a relapsing cutaneous disease caused by gluten sensitivity and is characterized by severely pruritic papulovesicles or excoriated papules on the extensor surfaces, scalp, nuchal area, and buttocks. DH is considered an extraintestinal manifestation of celiac disease (CD). CD is an inflammatory disease of the small bowel also due to gluten sensitivity. DH is rare, with a reported prevalence between 11.2 to 75.3 per 100,000, while CD is much more common, with an estimated prevalence of 1400 per 100,000 [1,2,3,4]. They both share multiple features pertaining to pathogenesis, enteropathy findings, and treatment, but differ in various ways as well. This review aims to comprehensively describe DH and differentiate it from CD, with an emphasis on the current diagnostic methods, disease monitoring serologies, and management.

## 2. Epidemiology

DH has a reported incidence between 0.4 to 3.5 per 100,000 people per year and prevalence between 11.2 to 75.3 per 100,000 [1,2,3]. The higher rates are often found in countries such as Finland due to this disease’s predilection for individuals of northern European descent [2]. Conversely, DH is rare among Asian populations and even rarer among African Americans [1]. DH can occur at any age, but is most commonly diagnosed between 30 to 40 years of age, with a mean of 43 years. There is a male predominance with a male to female ratio between 1.5:1–2:1 [3].

## 3. Pathogenesis

The pathogenesis of DH is similar to that of CD, as both are complex, involving interactions among genetic, immunologic, and environmental factors. Gluten hypersensitivity has a strong genetic component as first-degree relatives of both DH and CD patients have an almost 15-fold increased risk compared to the general population [5]. Both DH and CD are closely associated with human leukocyte antigen (HLA) DQ2 and DQ8 haplotypes; up to 90% of cases are associated with HLA DQ2 and the remainder with HLA DQ8 [6,7,8]. They are both involved in the processing of the gluten antigen gliadin. The immunologic reactions that underlie the pathogenesis of CD is initially similar in DH. Tissue transglutaminase (TG2/tTG), which is present in the gut, is the main autoantigen in CD. TG2 modifies glutamine to glutamic acid within gliadin, which is an alcohol-soluble fraction of gluten, after gliadin is absorbed in the lamina propria of the gastrointestinal (GI) lumen. This modification is the critical step that causes gliadin to have a stronger affinity for HLA DQ2 and DQ8 on antigen presenting cells. Subsequent presentation of gliadin to CD4+ T-cells results in inflammation and mucosal epithelial cell damage. The modified glutamine residues of gliadin also cross-link covalently to TG2, and present to gliadin-specific helper T-cells, which then stimulate B-cells to produce circulating IgA antibodies directed against TG2. By epitope spreading, circulating IgA class autoantibodies also form against epidermal transglutaminase (TG3/eTG) found in the skin. TG3 is the main autoantigen in DH, as opposed to TG2 in CD. The pathogenesis of DH differs from CD as high-affinity anti-TG3 antibodies deposit in the dermal papillae and form a complex with TG3 produced by keratinocytes; this triggers a local inflammatory response within the papillary dermis that is predominantly neutrophilic. It is proposed that DH starts with hidden CD as a TG2 immune response in the gut which evolves into a TG3 response in the papillary dermis as a late manifestation of CD. Of note, both patients with CD and DH produce anti-TG3 antibodies, but in CD these have low affinity for TG3 and thus do not form the immune complexes that deposit within dermal papillae as opposed to the high affinity anti-TG3 antibodies found in DH [9]. More research is needed to further elucidate this mechanism.

## 4. Clinical Features

The distribution and morphology of DH has hallmark features: DH follows a symmetric distribution and involves the extensor surfaces, such as the elbows, dorsal forearms, knees, and buttocks; Figure 1 depicts DH of the buttocks. Other regions that are often affected include the scalp, neck, upper back, and sacral region. The face and groin; however, can also be affected [1,10,11]. The eruption is usually polymorphic, consisting of groups of erythematous papules, urticarial plaques, and vesicles. Due to severe pruritus and subsequent scratching, many patients present with erosions, crusted papules, and excoriations which usually heals without scars [1,9,11]. Because pruritus is such a prominent feature, its absence strongly favors another diagnosis [9]. Although lesions have been described as vesicles, macules, and erosions in the oral mucosa, mucosal involvement is rare in DH. Uncommonly, petechiae and purpura, particularly on the palms and soles, can present alongside classic manifestations or as the sole presenting feature of DH [12,13,14]. Other uncommon presentations of DH include palmoplantar keratoses, urticarial plaques, and prurigo pigmentosa-like lesions [15,16,17,18]. Dental anomalies including enamel defects (enamel pits, horizontal grooves, defects in enamel color), and delayed eruption of teeth have also been reported in patients both in CD and DH [1].

Trichoscopy was studied to evaluate autoimmune bullous diseases on the scalp and can potentially be used to differentiate DH from other diseases of this class. On dermoscopy, DH displays extravasations (8/8, 100%), yellow hemorrhagic crusts (3/8, 37.5%), and characteristically clustered dotted vessels (5/8, 62.5%) and white diffuse scaling (4/8, 50%), but no yellow scale or well-demarcated creamy-yellow structures [19]. Given the small sample sizes and overlapping features among autoimmune bullous diseases, further study is need for validation.

### 4.1. Differential Diagnosis

The differential diagnosis of DH includes vesiculobullous diseases and pruritic diseases, both of which can share clinical traits and even histopathological findings with DH. The primary autoimmune bullous disorders that should be differentiated from DH are linear IgA disease, bullous pemphigoid, and epidermolysis bullosa acquisita. Pruritic diseases such as urticaria, atopic dermatitis, eczema, scabies, prurigo, and lichen planus may be confused clinically with DH as well. Clinically, the symmetric involvement of extensor surfaces may guide the diagnosis of DH. DH can ultimately be distinguished from these mimickers by the hallmark finding of IgA deposits in the dermal papillae and/or dermoepidermal junction on DIF. Serological autoantibodies such as anti-TG2 and anti-TG3 antibodies may also help practitioners identify DH among these other entities [1,11,20].

### 4.2. Enteropathy

Most patients with DH have evidence of some degree of celiac-type damage in their small bowel; however, this is usually milder than CD. Consequently, patients with DH may have a history of GI symptoms such as bloating, diarrhea, or constipation, but these are usually minor if present. Due to the association with gluten-sensitive enteropathy, patients with DH can present with associated complications of malabsorption such as nutritional deficiencies, osteoporosis, short stature, anemia, and weight loss [21,22,23,24]. These are rare in DH; however, unlike in CD [18]. They can also develop celiac-related complications such as celiac sprue, ulcerative ileitis, or non-Hodgkin lymphoma as well as GI malignancies [21]. The diagnosis of DH is associated with a significantly increased risk of non-Hodgkin lymphoma as in CD. However, this risk is increased only during the first five years after diagnosis. Both T-cell and B-cell lymphomas have been reported, with B-cell lymphomas occurring more commonly [25,26].

### 4.3. Neurologic Dysfunction

Rarely, gluten sensitivity has been associated with neurologic dysfunction such as cerebellar ataxia, polyneuropathies, epilepsy, myelopathy, and encephalopathy [27,28]. Case reports have described DH accompanied by various neurologic pathologies [29,30]. The incidence rates of these diseases have not been studied in DH but are presumably low. Nonetheless, dermatologists should be aware of this association and refer to a neurologist when necessary.

### 4.4. Autoimmune Diseases and Associated Conditions

There are associations between the diagnosis of DH and a myriad of other autoimmune diseases, with the most common being autoimmune thyroid disease and type I diabetes mellitus. Other infrequently reported conditions include pernicious anemia, multiple sclerosis, Sjögren syndrome, SLE, rheumatoid arthritis, vitiligo, alopecia areata, dermatomyositis, sarcoidosis, Addison disease, psoriasis, and atopic dermatitis [11,21,22,31]. Most clinicians can easily screen patients with DH for thyroid disease (TSH, T3, T4, and anti-thyroid peroxidase) and type 1 diabetes (serum glucose) and should test for other autoimmune conditions based on associated signs and symptoms. DH is also associated with an increased risk of bullous pemphigoid. The diagnosis of DH and the subsequent diagnosis BP have been reported with variable intervals. Therefore, dermatologists should be cognizant of the possibility of the new diagnosis of bullous pemphigoid when clinical presentations change by the formation of large bullae and/or gluten free diet is no longer effective [32].

## 5. Diagnosis

The diagnosis of DH is based on a consistent clinical picture coupled with serology, immunofluorescence, and histopathology. Direct immunofluorescence (DIF) remains the gold standard for diagnosis, but histopathological and serological tests are used as adjuncts to further aid diagnosis [33].

### 5.1. Serology

Various serologic tests can be used as adjunct for the diagnosis of DH in equivocal cases. Most use the detection of autoantibodies, many of which are also used in the diagnosis of CD.

As mentioned in pathogenesis, TG3 (eTG) is the major autoantigen in DH [34]. Previously, the enzyme-linked immunosorbent assay (ELISA) for IgA antibodies to TG3, was not as widely available and used primarily for research purposes [21]. However, this assay has been commercially available in the US for almost a decade. The sensitivity of this test has been reported between 52% and 100% [21]. In 2021, Betz et al. reported anti-TG3 antibody detection was more sensitive in diagnosing DH than antibodies to TG2 and endomysium (EMA). In fact, 38% of their biopsy proven DH patients were negative for both TG2 and EMA [35]. It is rare for a DH patient to be positive for anti-TG2 antibodies and negative for anti-TG3 antibodies [35,36]. Anti-TG3 antibodies are also more specific for DH as compared to CD with a reported specificity between 90% and 100% [21]. Borroni et al. showed that anti-TG3 antibody serum levels were higher in DH patients than in CD patients without DH. They also showed that anti-TG3 antibody levels could distinguish untreated DH from other pruritic skin diseases [37]. Although celiac patients without skin disease and celiac patients with skin diseases other than DH still produce anti-TG3 antibodies, only in DH are the antibodies high affinity. Therefore, the authors prefer anti-TG3 antibodies, when available, as the first line serologic marker for diagnosis of DH, alongside DIF. In addition, this antibody level correlates with disease activity and the extent of small bowel damage in DH patients more than other antibody levels [38].

The assay for IgA autoantibodies directed against TG2 (tTG) is the most often used to diagnose CD. This test is commonly used as it is widely available, inexpensive, and easy to perform. ELISA for IgA class antibodies to TG2 has a high sensitivity for both DH and CD, with a reported sensitivity for DH as high as 95% [21,39]. Though highly sensitive, anti-TG2 antibodies are not as specific to DH as anti-TG3 antibodies; anti-TG2 antibodies are also found in CD patients without DH and CD patients with skin diseases other than DH. Serum anti-TG2 antibodies decrease to normal levels in patients on a GFD and increase while relapsing, and thus a preferred test for monitoring compliance with the GFD in DH patients. However, in about 20% of DH patients, IgA anti-TG2 antibody levels are negative despite gluten exposure, but positive for IgA anti-TG3; in these cases, anti-TG3 is used [36]. TG2 is also the target for endomysial antibodies.

IgA antibodies to endomysium (EMA) are directed against smooth muscle reticular connective tissue of the esophagus, stomach, and small intestine. They are detected by indirect immunofluorescence (IIF) on monkey esophagus and are commonly used in the diagnosis of CD. Due to its cost and operator-dependent nature, IIF for anti-EMA antibodies is typically limited to more complex or difficult-to-diagnose cases [21,39]. Of note, IIF with DH serum using normal human skin as a substrate is routinely negative for circulating IgA antibodies, failing to reproduce the pattern of granular IgA deposition in dermal papilla. This is due to the lack of autoantigen–antibody complex in the dermal papillae of normal skin. Therefore, IIF should not be ordered on the skin for the diagnostic workup of DH.

IgA- and IgG-class antibodies to deamidated gliadin peptides (DGP) are other markers used in CD that are not as well studied in DH. Studies show sensitivities for DH patients below that of anti-TG2 antibodies, ranging from 66–72% [40]. Anti-DGP antibodies are also reserved for unclear clinical pictures.

Total IgA level is usually examined as well. Selective IgA deficiency has not been reported with DH, in contrast to CD, but partial IgA deficiency has been reported. In these select cases, testing for levels of IgG antibodies to transglutaminase and EMA is useful [41]. Otherwise, testing should be limited to IgA class autoantibodies.

Serological tests pertaining to DH are still in development. In a 2021 study, Ziberna et al. created a novel ELISA for measuring anti-TG3 antibodies with a high diagnostic performance [42]. Additionally, in 2021, a new bi-analyte immunoblot test detecting IgA to both TG2 and gliadin simultaneously was effective in diagnosing DH [43]. Other promising antibodies that are under investigation include anti-TG6, anti-neoepitope TG2, and anti-GAF3X [40,44,45]. Interleukin-36 (IL-36) levels were found to be significantly elevated in DH as compared to other autoimmune bullous diseases [46]. Further testing is required to prove their utility in the diagnostic workup of DH.

### 5.2. Direct Immunofluorescence

DIF is the gold standard test for diagnosis of DH, with a sensitivity of 90–95% and specificity of 95–100% [23,24,47]. Biopsy should be obtained from uninvolved perilesional skin, as this lesion contains significantly greater number of IgA deposits, and lesional biopsies in general have a higher false negative rate [11,48]. The pathognomonic finding on DIF is granular IgA deposits in the dermal papillae and/or dermoepidermal junction [47,48,49]. However, DH with the finding of IgA deposits solely along the dermoepidermal junction may be confused with linear IgA bullous dermatosis, in which case further serologic testing, as described below, is required to differentiate the two [49]. Less frequently, a fibrillar pattern of IgA deposits has been described in the dermal tips of DH patients. This pattern differs from the granular pattern in that the IgA deposition appears as linear streaks rather than fine granules [50]. This pattern is more common in Japan and can be found in up to 50% of patients there [51]. In addition to IgA, IgM and C3 have also been identified at the dermal papillae and dermoepidermal junction [9].

If a patient with high clinical suspicion for DH displays a negative DIF result, clinicians should consider repeating the exam with a biopsy from a new site of normal appearing perilesional skin [21]. False-negative results occur in about 5% of biopsies [48]. A strict GFD can decrease IgA levels in the skin, also affecting DIF results in contrast to pharmacologic treatments which do not alter IgA deposits. Providers should evaluate the patient’s diet, and if the patient is on a GFD, the patient should be re-biopsied after one month of consuming a normal diet. In rare instances, repeat DIF examination may be negative in DH patients [52]. In this case, the combination of clinical, histopathological, and serologic data can be used to form the diagnosis [21].

### 5.3. Histopathology

A lesional skin biopsy of an entire intact vesicle is preferred for hematoxylin and eosin (H&E) staining. If no intact vesicles can be found, the biopsy should be taken from intact, erythematous skin [37]. The typical histopathological findings of DH are subepidermal vesicles and blisters with an accumulation of neutrophils at the tips of dermal papillae (papillary microabscesses) with relative sparing of the lower tips of rete ridges as seen in [53] (Figure 2). Histopathology is not necessary for the diagnosis of DH. These findings are not specific for DH as linear IgA disease and bullous systemic lupus erythematosus can present with identical histopathologic findings [54]. Occasionally, eosinophils may infiltrate the papillary tips as well [11]. The findings by routine light microscopy may be nonspecific in about one-third of DH cases, exhibiting perivascular lymphocytic infiltrate and minimal papillary inflammatory infiltrate [55].

### 5.4. HLA Testing

Almost all DH and CD patients have either HLA DQ2 (90%) or DQ8 alleles (10%). Therefore, testing for HLA DQ2 or DQ8 haplotypes has a high negative predictive value, and a negative test can be used to exclude DH as a diagnosis. Because of high prevalence of these alleles in the general population (30–40%), and consequently a low specificity for DH, a positive test is not helpful to diagnose DH. Therefore, genetic testing is not recommended for routine workup of DH [21,47].

### 5.5. Small Bowel Biopsy

Although most DH patients have minimal GI symptoms, a large majority, up to 90%, have some degree of enteropathy, from villous atrophy and crypt hyperplasia to mild intraepithelial lymphocytes [11,18]. Unlike in CD, small bowel biopsy is not required for the diagnosis of DH, as DIF and serology both have high sensitivity and specificity [24]. In addition, because DH is considered an extraintestinal manifestation of CD, once DH is diagnosed, small bowel biopsy does not need to be performed [11,23]. Recently, other authors have strongly recommended small bowel biopsy in most DH patients to assess the degree of enteropathy [20]. However, the degree of small bowel damage at diagnosis does not affect the long-term prognosis of DH [56]. To spare the patient from this invasive procedure, we reserve small bowel biopsy if DH cannot be diagnosed for suspected cases, if severe GI symptoms incongruent with typical CD are present, or if the clinician suspects GI malignancy including lymphoma [11,21,57].

### 5.6. Screening Family Members

Hervonen et al. reported that 18% of patients with DH had first-degree relatives with CD or DH and a prevalence of 5.4% of first-degree relatives of DH had CD or DH [5]. Due to increased number of affected individuals with DH or CD among the family members of patients with DH, some authors advocate serologic testing for CD and DH among first- and second-degree family members [18].

### 5.7. Diagnostic Approach

Based on the current evidence as described previously, our recommended diagnostic algorithm, including tests and considerations following diagnosis of DH, is outlined (Figure 3). To reiterate, in contrast to other authors, we prefer anti-TG3 antibodies, when available, over anti-TG2 antibodies in the diagnosis of DH due to its superior specificity [20]. We use anti-TG2 antibodies as a primary diagnostic tool if anti-TG3 testing is unavailable, as a secondary diagnostic tool in unclear cases, and for dietary compliance monitoring. Practitioners should also be aware that a higher diagnostic delay was observed in males compared to females, potentially due to negative serologies occurring more commonly in males [58].

## 6. Disease Monitoring

Other than the visual and symptomatic improvement of the rash, serologic tests can be used to monitor DH patients’ adherence and response to treatment. Serum IgA antibodies against TG2, TG3, and EMA can be used to monitor activity of disease and are related to the degree of adherence to the diet, while IgA and IgG levels of DGP were not. Anti-TG2 and anti-TG3 antibodies are the most commonly used markers for disease monitoring of DH at follow-up visits given the convenience of ELISA, as compared to IIF for EMA antibodies. ELISA for anti-TG3 antibody correlates with cutaneous manifestations as well as small bowel damage in DH with high accuracy, arguing against the need for routine small bowel biopsies in these patients. Thus, anti-TG3 antibody is our preferred marker for monitoring disease activity in patients with DH. Additionally, ELISA for anti-TG2 antibodies is more sensitive than anti-EMA and anti-DGP antibodies in detecting gluten exposure. As such, anti-TG2 antibodies are our preferred marker for monitoring dietary compliance in DH patients. In around 20% of DH patients, however, anti-TG2 antibody levels are negative despite gluten exposure, in which case anti-TG3 antibodies are a useful substitute [36,57,59].

## 7. Management

A lifelong gluten-free diet (GFD) is the first-line treatment for DH and CD. Strict adherence to GFD will lead to resolution of skin and bowel diseases. GFD monotherapy typically takes a few months and up to several years to achieve remission in DH [18]. Consequently, during the first 1–2 years after diagnosis and/or during “flare-ups” of the disease, drugs such as dapsone, other sulfonamides, or steroids can be useful as short-term additive treatments until diet alone is adequate [9]. However, not all cases can be controlled with this regimen, and clinicians should be informed of the alternative medications being used for recalcitrant cases, summarized in Table 1. Case reports have documented successful use of uncommon medications in select DH cases and require further evaluation to assess effectiveness.

Management of DH requires close follow-up from a dermatologist, gastroenterologist, and dietician. All these providers are needed to routinely assess the patient’s diet adherence, response to treatment, medication side effects, and possible development of complications. The multidisciplinary team may also include an internist, rheumatologist, or neurologist, depending on the associated comorbidities and complications [24].

### 7.1. Gluten-Free Diet

Like CD, adherence to a strict GFD is the mainstay of treatment in DH. Patients are advised to avoid any gluten containing foods made from cereals including wheat, barley, rye, and malt. The FDA defines gluten-free foods as those containing <20 ppm of gluten, though in other countries, products with <100 ppm may carry the label [9]. Of note, pure oat products can be consumed by DH patients. A recent study found oats to not only be safe in DH patients, but in the long term may improve quality of life and GI symptoms [60]. However, DH patients should be cautious as most store-bought oat products are typically contaminated with gluten, therefore it is recommended to avoid those oats or oats containing products [9]. Other examples of gluten-free foods that are safe to eat include whole-grain rice, maize, potatoes, and vegetables [24].

The effects of a GFD on DH and CD severity are significant. In 133 DH patients studied by Garioch et al., advantages of strict dietary management included reduced need for medication, resolution of enteropathy, increased feeling of well-being, and a protective effect against lymphoma [61].

Adherence to GFD will resolve cutaneous and GI symptoms of DH, although GI symptoms often respond faster than the skin disease. A lifelong GFD will help achieve optimal disease control, avoidance of complications, and even full resolution after a mean of 2 years. However, adherence is difficult in practice because it requires meticulous monitoring of food labels and ingestion, can be costly and inconvenient, and is socially limiting [62]. Dieticians and support groups are helpful in navigating the challenges of adhering to this diet and finding hidden sources of gluten. Patient and family communication should emphasize the importance of diet adherence even in the absence of GI manifestations, as well as the chronicity of the disease and its management.

Within the past two decades, several studies have examined the possibility of long-term remission of DH in 10–20% of cases, suggesting the possibility of discontinuation of GFD in well-controlled patients [63,64]. However, a recent study found that 95% (18/19) of DH patients that were well controlled on long term GFD, relapsed with a gluten challenge [65]. Thus, for now, we recommend lifelong GFD, and more research is needed to determine the safety of returning to a gluten-containing diet. Lastly, a small percentage (1.7%) of DH patients have a rash that is not responsive to GFD monotherapy for a mean of 16 years, coined “refractory DH”, and require supplementary treatment (dapsone) for cutaneous improvement [66].

### 7.2. Dapsone

The primary drug used in DH treatment is dapsone, a sulfonamide that has both anti-inflammatory and antibacterial properties. It is most useful in treating diseases with neutrophilic infiltrates. It was demonstrated that dapsone was able to inhibit the myeloperoxidase-peroxide-halide-mediated cytotoxic system and neutrophil respiratory burst, which may control the degree of neutrophil-induced destruction. Also, dapsone reduces hydrogen peroxide and hydroxyl radical levels, both strong scavengers of reactive oxygen species, and reduces eosinophil-mediated tissue damage [67,68].

Dapsone works quickly, resolving symptoms of pruritis in several hours and new blister formation in 24–36 h, making it an effective option for acute inflammatory phases of disease as well as early stages of GFD initiation. Please note that DH symptoms and rash usually return within 24 to 48 h of halting medication if the DH has not yet been adequately controlled without medication. Dapsone has no effect on the enteropathy, IgA deposition, or lymphoma risk [23].

There are different approaches for dosing dapsone. One approach is to start 25–50 mg/day in adult patients, which can be gradually increased to 100–200 mg/day if needed. Another approach, if the patient does not have any significant risk factors including severe cardiac, pulmonary, or hematologic disease, is to start with 100 mg/day, which can control the disease rapidly in most DH patients. The dose can then be adjusted to achieve the lowest possible amount needed to control the disease [68]. Most DH patients can be managed with 100–200 mg of dapsone daily, although doses range from 25–400 mg/day. Unsurprisingly, Jarmila et al. observed that increasing dapsone dose was associated with DH disease severity [69]. Regardless of different dosing methods, dapsone can be slowly tapered off following resolution of the rash; on average, this takes approximately 2 years on a strict gluten-free diet and can take longer with incomplete diet adherence [70,71].

Dapsone is a well-tolerated medication in DH patients [69]. However, dose-dependent hemolysis and methemoglobinemia are well recognized side effects, particularly in those with glucose-6-phosphate dehydrogenase (G6PD) deficiency. Hemolytic anemia occurs in all individuals to some degree. A 2021 study investigating the prevalence of anemia in DH found that although anemia in DH is relatively low, it is more prevalent when the patient is using dapsone [72]. Methemoglobinemia occurs when red blood cells contain methemoglobin at levels higher than 1%. Methemoglobin is formed when the heme iron is converted to the ferric form (Fe^3+^) instead of the ferrous form (Fe^2+^), resulting in decreased oxygen delivery to tissues. Clinical symptoms are proportional to the amount of methemoglobin in the blood, ranging from lightheadedness and fatigue to respiratory depression, coma, and even death. Other less frequent side effects of dapsone include headache, general malaise, nausea, elevated transaminases, peripheral neuropathy (primarily distal motor neuropathy with some sensory involvement), agranulocytosis, and dapsone hypersensitivity syndrome [67,68]. Agranulocytosis (typically occurring 3–12 weeks after starting the medication) and dapsone hypersensitivity (usually 4–6 weeks after initiating treatment with incidence of 0.5 to 3.5%) are rare but serious complications and dapsone should be discontinued immediately. Both are idiosyncratic adverse reactions. Thus, clinicians should perform close and regular laboratory monitoring of labs. At baseline, a complete blood count (CBC) with differential, liver function tests (LFTs), renal function tests, and G6PD level should be checked. Then, CBC should be done weekly for the first month then every 2 weeks for the next 8 weeks and every 3–4 months thereafter. LFTs should be monitored every 2wks for the first month, then checked every 3–4 months along with renal function. Methemoglobin levels or reticulocyte count can be checked if methemoglobinemia or hemolytic anemia is suspected, respectively [47,73]. The presence of cyanosis, failure of hypoxemia to resolve with supplemental oxygen, or a substantially lower pulse oximetry reading are all clues that raise suspicion of methemoglobinemia [74]. Supplementation with cimetidine or high dose vitamin C have been demonstrated to reduce the risk of methemoglobinemia. Vitamin E has also been shown to prevent methemoglobin formation and hemolysis [22]. If methemoglobinemia does occur, dapsone should be discontinued and oral methylene blue, along with supportive care, should be considered for rescue. Methylene blue should be avoided in patients with G6PD deficiency as it can cause severe hemolysis, in which case vitamin C should be given instead [75]. If oral dapsone and its alternatives are not an option, topical dapsone 5% gel may be considered for mild diseases, as it is free of systemic side effects and appears to be moderately effective at treating localized face and chest lesions [10,76].

### 7.3. Sulfonamides Other Than Dapsone

If the patient cannot tolerate due to adverse effects or does not respond to dapsone, other drugs in the sulfonamide class may be used. Previously, sulfapyridine (1–2 g/day) and sulfamethoxypyridazine (0.25–1.5 g/day) were used, but sulfamethoxypyridazine is no longer available and sulfapyridine can be obtained through compounding pharmacies. Currently, sulfasalazine (1–2 g/day) is the only one commercially available in US [21,77,78]. They do not cause hemolysis seen in dapsone. Their most common side effect is GI upset, but less frequently may cause hematologic toxicities (neutropenia, agranulocytosis, thrombocytopenia, aplastic anemia in the first 1–3 months of starting the medication). Proteinuria and crystalluria are also possible side effects. Routine lab monitoring of CBC with differential and UA is required with these drugs, albeit less frequently than with dapsone (monthly for the first 3 months, followed by every 6 months). Patients should also maintain adequate hydration during treatment to prevent crystalluria [77]. These may also be used in combination with dapsone to achieve a more complete response [79].

### 7.4. Other Medications

Other medications such as superpotent or potent topical steroids can also be considered for acute symptomatic relief and reduction of localized lesions in DH [33]. Examples of useful topical steroids include betamethasone valerate, dipropionate, or clobetasol propionate. In contrast, systemic corticosteroids are not effective treatments for DH. Pruritus associated with DH has shown poor response to systemic steroids and antihistamines [47]. As previously mentioned, topical 5% dapsone is another viable option that has been shown to be moderately effective for localized disease [10,76]. Neither topical steroids nor topical 5% dapsone should be used as monotherapy but should instead be used for local disease control during acute flares alongside a systemic treatment.

Treatment of DH becomes challenging if GFD and sulfonamides are inadequate due to intolerable side effects, contraindications, and/or inefficacy. Case reports and series have documented the benefit of alternative medications such as methotrexate, colchicine, cyclosporin, heparin, tetracycline, mycophenolate, azathioprine, nicotinamide, and rituximab in DH [80,81,82,83,84,85,86]. Interestingly, colchicine inhibits neutrophil function and can be used in combination with a sulfonamide, usually reducing the sulfonamide dosage needed for treatment. On the other hand, cyclosporine may not be useful as the therapeutic dose needed may be harmful to the patient [20]. In addition, as rituximab has been shown to be effective in a recalcitrant case of DH, other biologics may be of therapeutic use as well, but further investigation is needed. Clinicians should exhibit caution with biologics, however, as infliximab has been reported to trigger DH [87].

IL–1, IL–17, and IL–36 have all been implicated in the pathogenesis of DH and may someday serve as novel targets for therapeutics [46,88].

## 8. Prognosis

DH is a chronic, remitting disease that is likely to be a lifelong condition in those who are diagnosed. However, patients with DH adhering to a strict GFD have an excellent prognosis. In fact, the quality of life of long-term GFD-treated patients with DH is comparable to the general population [89]. Despite the increased lymphoma mortality in the first 5 years of follow-up, patients with DH overall have a lower mortality rate compared to that of the general population when on a GFD [26,90]. Patients with DH are significantly less likely to have hypercholesterolemia or a smoking history than the general population [90]. Although it is possible to return to a normal diet without relapse in 20% of cases, the effects of this change on long term morbidity and mortality are not known [63,64]. As of now, there are no guidelines for transitioning well-controlled patients to normal diets, and it is prudent for all patients with DH to maintain a lifelong GFD to achieve optimal prognosis [11].

## 9. Conclusions

DH is an autoimmune blistering disease that is a manifestation of dietary gluten sensitivity. As it has been for many years, it is diagnosed primarily through DIF, in which IgA deposition in the dermal papillae is essentially pathognomonic. However, our modern approach uses anti-TG3, instead of anti-TG2, antibody levels as the primary serological diagnostic marker, when available. In equivocal cases, anti- TG2, anti-EMA, anti-DGP antibodies can aid diagnosis. Small bowel biopsy is rarely necessary for diagnosis or disease monitoring, unlike in CD. Following or concurrent with diagnosis, clinicians should assess patients for malabsorption and associated autoimmune diseases. Treatment typically involves a multidisciplinary team of a dermatologist, gastroenterologist, and dietician. Anti-TG3 and anti-TG2 antibodies are also valuable markers for disease activity and dietary compliance, respectively. Contemporary studies confirm strict, long-term GFD as the primary treatment modality. The diet is supplemented with medications with sulfonamides as first line treatment, especially dapsone. More recent studies have suggested that the use of biologics, such as rituximab, can be effective in recalcitrant cases. Interestingly, DH patients have been shown to exhibit lower mortality than the general population when well-managed. Finally, while closely related to CD, DH is not as well studied and thus requires further research to better understand specific nuances in diagnosis and management between the two.

## Figures and Tables

**Figure 1 medicina-57-00843-f001:**
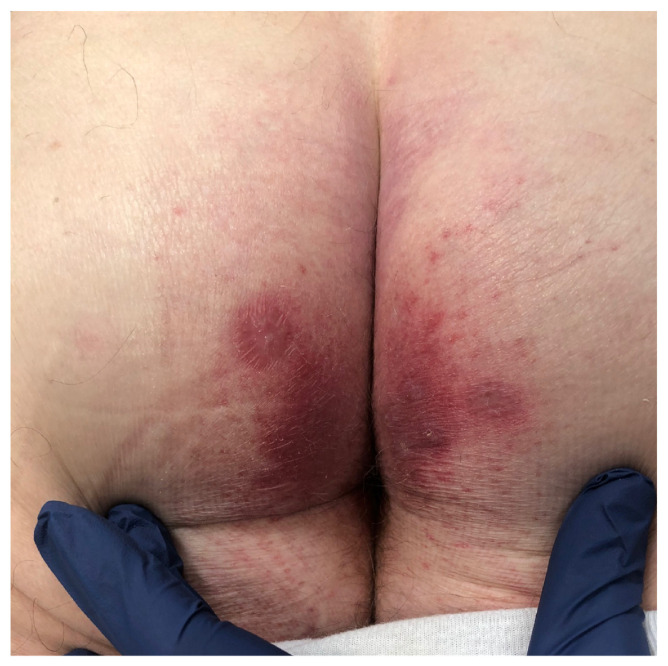
Clinical presentation of dermatitis herpetiformis (DH) on the buttocks: erythematous grouped papules and vesicles.

**Figure 2 medicina-57-00843-f002:**
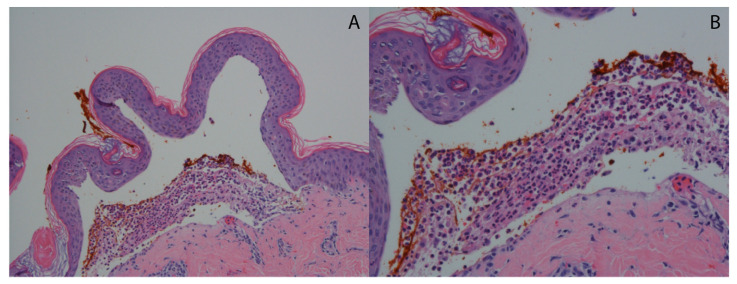
Typical histopathological findings in DH. (**A**) Hematoxylin and eosin-stained sample showing subepidermal separation (H&E 10×). (**B**) Hematoxylin and eosin-stained sample showing dense accumulation of neutrophils at the papillary dermis forming a microabscess (H&E 20×).

**Figure 3 medicina-57-00843-f003:**
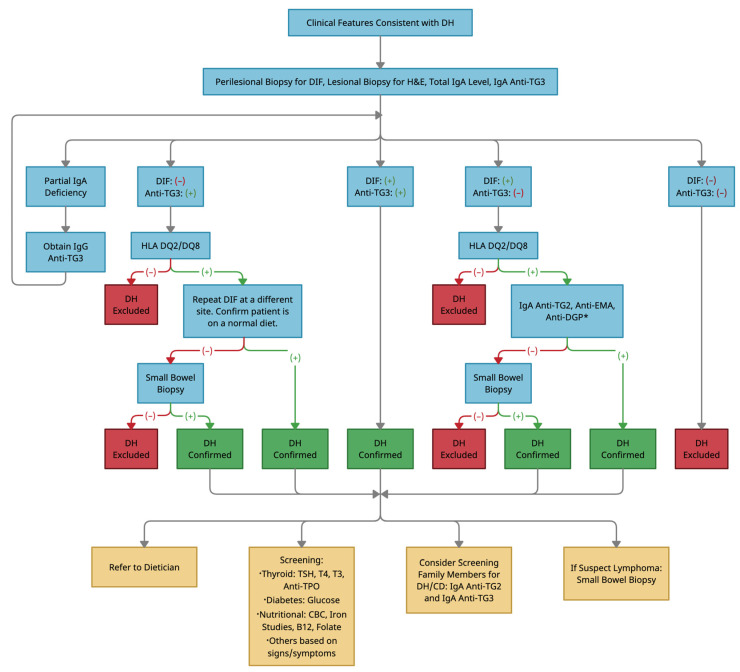
Diagnostic algorithm for a rash suspected of dermatitis herpetiformis. Abbreviations: DH, dermatitis herpetiformis; DIF, direct immunofluorescence; H&E, hematoxylin and eosin; IgA, immunoglobulin A; Anti-TG3, anti-epidermal transglutaminase antibodies; IgG, immunoglobulin G; HLA, human leukocyte antigen; Anti-TG2, anti-tissue transglutaminase antibodies; Anti-EMA, anti-endomysial antibodies; Anti-DGP, anti-deamidated gliadin peptide antibodies; TSH, thyroid stimulating hormone; T4, thyroxine; T3, triiodothyronine; Anti-TPO, anti-thyroid peroxidase antibodies; CBC, complete blood count; CD, celiac disease. * denotes obtaining IgG antibodies instead of IgA antibodies in the case of partial IgA deficiency.

**Table 1 medicina-57-00843-t001:** Dermatitis herpetiformis management options.

Medication	Dose	Remarks
**First Class**		
Gluten Free Diet	Not applicable	Strict, lifelong
Dapsone	25–400 mg/day	
**Second Class**		
Sulfasalazine	1–2 g/day	
Sulfapyridine	1–2 g/day	No longer available in US
Sulfamethoxypyridazine	0.25–1.5 g/day	Through compounding pharmacies
**Topical Adjuncts**		For local disease control during a flare
Topical Steroids	Various	
Topical 5% Dapsone	BID	Effective for primary facial involvement
**Alternatives**		Efficacy shown in select case series and reports
Methotrexate	5–25 mg/week	
Colchicine	0.6–2.4 mg/day	
Cyclosporine	3–7 mg/kg/day	Therapeutic dose may be in dangerous range
Heparin	500–1000 U/hr IV or 40 IU SQ	May be given with tetracycline and nicotinamide
Tetracycline	0.5–2 g/day	
Nicotinamide	0.1–1.5 g/day	
Mycophenolate	1 g/day	
Azathioprine	1–2.5 mg/kg/day	
Rituximab	375 mg/m^2^	Weekly dose for 4 weeks

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
