# Peer review of "Dermatitis Herpetiformis: An Update on Diagnosis, Disease Monitoring, and Management"

_medicina, 2021, doi:10.3390/medicina57080843_

Round 1
Reviewer 1 Report
This up-to-date review well described clinical features, diagnosis, management and prognosis of Dermatitis Herpetiformis reported in the literature.
Author Response
There are no comments needing a response. Thank you for the review.
Reviewer 2 Report
The authors' review of dermatitis herpetiformis is neither novel nor does it provide any new information. There are recent reviews on the subject (1). However, the review is acceptably well done and covers all the important aspects of the pathology.
1: Reunala T, Hervonen K, Salmi T. Dermatitis Herpetiformis: An Update on Diagnosis and Management. Am J Clin Dermatol. 2021 May;22(3):329-338. doi:10.1007/s40257-020-00584-2.
1.-In a review, a critical aspect to assess its quality is the bibliography reviewed and how it is presented in the text. The present manuscript requires a complete revision of the bibliography.
*There are references in the bibliography that are not cited in the text (9, 42, 24.......).
*Many of them are not related to the quoted text (30.......).
*The references are not presented in order, some being cited after others with subsequent numbers (41 , 43.....).
*In many cases, references are cited that have little or no relation to the quoted text and others that are directly related to the subject are omitted.
("As mentioned in pathogenesis, TG3 (eTG) is the major autoantigen in DH [44]."Here the most appropriate citation is 42, which is not cited in the text.)
("Of note, IIF with DH serum using normal human skin as a substrate is routinely negative
for circulating IgA antibodies, failing to reproduce the pattern of granular IgA deposition in dermal papilla. This is due to the lack of autoantigen-antibody complex in the dermal papillae
of normal skin [45]." Reference 45 has no relationship to this statement.).
2.-In the diagnostic algorithm for DH, the authors suggest the detection of anti-TG3 antibodies when this determination is currently very poorly developed. The algorithm proposed in ref 18 of Antiga et al seems more appropriate.
Reviewer 3 Report
August 6, 2021
Manuscript ID: medicina-1325025
Type of manuscript: Review
Title: Dermatitis Herpetiformis: A 2021 Update on the Review of Diagnosis and
Management
Opinion
The authors have done fantastic work. They have comprehensively described the Dermatitis Herpetiformis (DH) and have given an update until 2021. However, the manuscript is lacking when it comes to the ‘update’ term. I believe, the authors should work on this matter. I have some positive comments for them.
Comments
Comment#1: The title of the manuscript suggests it is an update about DH until 2021. However, the authors have provided mostly already known information. Does it mean there is no new addition in last years? The authors should give a heading in the manuscript describing what is new in this article? Otherwise, the title wouldn’t be justified. I see the authors have discussed various pharmacological agents for DH. They could be mentioned as updated information. The ‘conclusions’ section contains almost no updated information. I suggest trimming the lines providing already known information and add some updated information.
Comment#2: In the current form, the manuscript is not describing the aim of the study. The introduction looks quite incomplete and short. I suggest, in the introduction section, please describe briefly (a) what is DH? (b) what is celiac disease (CD)? (c) prevalence of DH, CD, and CD with DH, (d) How they are associated (just in 4-5 lines). Then provide the aim of this study and please mention at the end of this section that these entities are comprehensively described in the following sections.
Comment#3 (Pathogenesis): I agree that the pathogenesis of DH is complex and quite similar to CD. I suggest making clear the overlapping pathophysiological steps of CD and DH then mention the steps that are specific to DH only (eg. the involvement of TG3)
Comment#4 (Figure 1): If the authors have mentioned different surfaces why have provided they only the picture of one part? Either provide an image for all the mentioned parts or specify that the figure 1 shows the affected buttock surface. The way the Figure legend is given.
Comment#5 (Differential diagnosis): In the first view, this section looks quite incomplete. On further reading, I found an elegant explanation of the diagnosis of DH. I suggest the authors giving a summary of what the next sections are summarizing. This will not confuse the readers.
Comment#6 (Diagnosis): Please follow an order (start with serology then consecutive diagnosis options).
Comment#7 (Figure 3): This scheme is quite useful. I suggest providing a colored diagram. The image looks hazy please also provide a clear version.
Comment#1 (Section 5.2): This section is a vital and unique part of the manuscript. Please give a broad heading then summarize the current drug approaches of DH.

Round 2
Reviewer 2 Report
The manuscript has been suitably modified, improving its clarity. The bibliography has been corrected and new citations have been added.There is an abuse of citing other reviews instead of the original papers, it would be convenient and ethical to cite the original papers that have provided the knowledge.
Reviewer 3 Report
I agree with the responses. I don't have further comments. However, I suggest authors to expand the Introduction part (main body).